

# Rainforest conversion to rubber and oil palm reduces abundance, biomass and diversity of canopy spiders

Daniel Ramos[1], Tamara R. Hartke[1], Damayanti Buchori[2,3], Nadine Dupérré[4], Purnama Hidayat[3], Mayanda Lia[3], Danilo Harms[4], Stefan Scheu[1,5] and Jochen Drescher[1]

[1] Department of Animal Ecology, J.-F. Blumenbach Institute for Zoology and Anthropology, University of Göttingen, Untere Karspüle, Göttingen, Germany
[2] Center for Transdisciplinary and Sustainability Sciences, IPB University, Bogor, West Java, Indonesia
[3] Department of Plant Protection, Faculty of Agriculture, IPB University Bogor, Bogor, West Java, Indonesia
[4] Center for Taxonomy and Morphology, Zoological Museum Hamburg, Leibnitz Institute for the Analysis of Biodiversity Change (LIB), Hamburg, Germany
[5] Center for Biodiversity and Sustainable Land Use, Georg-August Universität Göttingen, Göttingen, Germany

Corresponding author
Daniel Ramos, dramosg@gwdg.de

## ABSTRACT

Rainforest canopies, home to one of the most complex and diverse terrestrial arthropod communities, are threatened by conversion of rainforest into agricultural production systems. However, little is known about how predatory arthropod communities respond to such conversion. To address this, we compared canopy spider (Araneae) communities from lowland rainforest with those from three agricultural systems in Jambi Province, Sumatra, Indonesia, *i.e.*, jungle rubber (rubber agroforest) and monoculture plantations of rubber and oil palm. Using canopy fogging, we collected 10,676 spider specimens belonging to 36 families and 445 morphospecies. The four most abundant families (Salticidae $N = 2,043$, Oonopidae $N = 1,878$, Theridiidae $N = 1,533$ and Clubionidae $N = 1,188$) together comprised 62.2% of total individuals, while the four most speciose families, Salticidae (S = 87), Theridiidae (S = 83), Araneidae (S = 48) and Thomisidae (S = 39), contained 57.8% of all morphospecies identified. In lowland rainforest, average abundance, biomass and species richness of canopy spiders was at least twice as high as in rubber or oil palm plantations, with jungle rubber showing similar abundances as rainforest, and intermediate biomass and richness. Community composition of spiders was similar in rainforest and jungle rubber, but differed from rubber and oil palm, which also differed from each other. Canonical Correspondence Analysis showed that canopy openness, aboveground tree biomass and tree density together explained 18.2% of the variation in spider communities at family level. On a morphospecies level, vascular plant species richness and tree density significantly affected the community composition but explained only 6.8% of the variance. While abundance, biomass and diversity of spiders declined strongly with the conversion of rainforest into monoculture plantations of rubber and oil palm, we also found that a large proportion of the rainforest spider community can thrive in extensive agroforestry systems such as jungle rubber. Despite being very different from rainforest, the canopy spider communities in rubber and oil palm plantations may still play a vital role in the biological control of canopy herbivore species, thus contributing important ecosystem services. The components of tree and palm canopy structure identified

as major determinants of canopy spider communities may aid in decision-making processes toward establishing cash-crop plantation management systems which foster herbivore control by spiders.

## INTRODUCTION

Tropical rainforests are among the most diverse terrestrial ecosystems and provide many ecosystem services, such as weather regulation and carbon storage at local, regional and global scales (*Sodhi et al., 2010*; *Böhnert et al., 2016*; *Codato et al., 2019*; *Milheiras & Mace, 2019*). Worldwide, they are under threat due to extraction of timber and minerals, as well as conversion into agricultural land-use systems, such as cattle farms and production of soy beans and palm oil (*Rudel & Roper, 1997*; *Sodhi et al., 2004*; *Grau, Gasparri & Aide, 2005*; *Renó et al., 2011*; *Barber et al., 2014*; *Vijay et al., 2016*). Deforestation rates are very concerning in Southeast Asia (*Koh & Wilcove, 2008*), particularly Indonesia, which in 2012 experienced the highest deforestation rates worldwide (*Margono et al., 2014*). Among the large islands of Indonesia, Sumatra has experienced the highest deforestation rates in the last decades (*Miettinen, Shi & Liew, 2011*; *Margono et al., 2014*) but has recently been surpassed by Kalimantan (*BPS, 2019*). A potential cause is that the Sumatran lowlands are already largely converted to non-forest land-use systems, such as agriculture, settlements and mining, while this process is at an earlier stage in Kalimantan. In Jambi Province, Sumatra, plantations and non-forest shrub land (61.8%) cover more than twice the area of primary and secondary rainforest (29.7%) (*Melati, 2017*).

Rubber and oil palm cash crops have become an increasingly dominant factor in overall Indonesian agricultural output over the last decades (*BPS, 2019*). In Jambi Province, rubber and oil palm plantations covered almost 670,000 and 500,000 ha in 2017, respectively, equaling the area of remaining rainforest (*BPS, 2018*). Most remnant rainforests are located in the mountainous west of the province and in some mountainous national parks such as Bukit Duabelas and Bukit Tiga Puluh, with only small patches of rainforest in the lowlands.

Recent studies show that transformation of lowland rainforest into monocultures of rubber and oil palm leads to substantial losses in abundance, and functional and taxonomic diversity as well as compositional shifts across a wide range of animal and plant groups (*Barnes et al., 2014*; *Mumme et al., 2015*; *Böhnert et al., 2016*; *Prabowo et al., 2016*; *Rembold et al., 2017*; *Paoletti et al., 2018*; *Potapov et al., 2020*). Large mammals are the most conspicuous faunal group affected by rainforest loss (*Nyhus & Tilson, 2004*), but the most severe consequences of rainforest transformation are associated with arthropods, which contribute the overwhelming majority of terrestrial animal species (*Hamilton et al., 2010*; *May, 2010*) and biomass (*Bar On, Phillips & Milo, 2018*). Tropical rainforest canopies are inhabited by one of the most diverse arthropod faunas (*Dial et al., 2006*; *Basset et al., 2012*;

*Floren, Wetzel & Staab, 2014*), which are particularly susceptible to the conversion into plantation systems such as rubber and oil palm due to direct habitat loss (*Turner & Foster, 2009*; *Fayle et al., 2010*).

Spiders (Araneae) are among the top predators in the arthropod food web, feeding mainly on insects and occasionally other arthropods (*Nelson & Jackson, 2011*). Some spiders are also known to consume larger prey, such as earthworms (*Nyffeler et al., 2017*), small skinks (*Shine & Tamayo, 2016*), and even small amphibians, birds and mammals (*Nyffeler & Vetter, 2018*; *Babangenge et al., 2019*). It is estimated that 400–800 million tons of prey are killed by the global spider community each year (*Nyffeler & Birkhofer, 2017*). Many spiders are web-builders while others are free hunters, which sets them apart ecologically from other major arthropod predator groups, such as centipedes and predatory beetles, and allows analysis of data according to basic ecological and biological characteristics. In addition to their role as predators, spiders are prey to a number of invertebrates and vertebrates, notably other spiders, parasitoid wasps, lizards and birds (*Wise, 1993*). As such, spider abundance and diversity may have major effects on their environment, including the decomposer system (*Wise et al., 1999*; *El-Nabawy et al., 2016*) and agricultural pests (*Suenaga & Hamamura, 2015*; *Rana et al., 2016*). Tropical rainforest conversion to rubber and oil palm plantations may thus have cascading top-down and bottom-up effects through the entire food web, and is likely to shape ecosystem functions and services of the converted ecosystems (*Potapov et al., 2020*).

Here, we studied canopy spider abundance, biomass, richness and community composition across a land-use gradient from tropical lowland rainforest via "jungle rubber" (rubber agroforest; *Gouyon, de Foresta & Levang, 1993*) to monocultures of rubber and oil palm in Jambi Province, Sumatra, Indonesia (*Drescher et al., 2016*). Based on previous studies on other taxa at our study sites, including ants (*Nazarreta et al., 2020*; *Kreider et al., 2021*), salticid spiders (*Junggebauer et al., 2021*) and parasitoid wasps (*Azhar et al., 2022*) we hypothesized that (1) canopy spider abundance, biomass and richness declines from rainforest to jungle rubber to rubber to oil palm monocultures. We further hypothesized that (2) the community composition of canopy spiders differs among each of the land-use systems, with the exception of rainforest and jungle rubber, which we hypothesized to be similar due to comparable structural complexity of the canopies. Lastly, using a large dataset of environmental variables, we hypothesized that (3) changes in the structure of canopy spider communities are driven by changes in habitat structure and associated changes in climatic factors such as temperature and relative humidity.

## MATERIALS & METHODS

### Sampling

The study was carried out within and surrounding two rainforest reserves in Jambi Province, Sumatra: the Bukit Duabelas National Park (S 01°59′41.4″, E 102°45′08.5″) and Harapan Rainforest (S 02°09′52.9″, E 103°22′04.0″) (Fig. S1). The area surrounding these two reserves is dominated by agroforestry systems, predominantly cash crop monocultures of rubber and oil palm (*Drescher et al., 2016*), but also jungle rubber, an agroforestry

system in which rubber trees are planted in successively degraded rainforest (*Gouyon, de Foresta & Levang, 1993*; *Rembold et al., 2017*). Canopy arthropods were sampled from three target canopies in each of eight research plots per land-use system, *i.e.*, lowland rainforest, jungle rubber, rubber and oil palm (*Drescher et al., 2016*, Fig. S2). Using the Swingtec SN50 fogger, we applied 50 mL DECIS 25 (Bayer Crop Science; active ingredient deltamethrin, 25 g/L) dissolved in four liters petroleum white oil to each of the target canopies within the first hour after sunrise to avoid turbulences during the day. The three target canopies were randomly chosen to represent overall canopy structure in the plots, *i.e.*, canopy gaps and fallen trees were avoided. Underneath each target canopy, 16 collection traps measuring 1 m × 1 m were suspended from ropes attached to height-adjustable tent poles; each trap was fitted with a plastic bottle containing 100 mL of 96% EtOH (Fig. S3). Two hours after fogging, the collection traps of each target canopy were collected and stored at −20 °C for future use. Arthropods of all three sampled target canopies were later determined to order. The study was conducted based on Collection Permit No. S.710/KKH-2/2013 issued by the Ministry of Forestry (PHKA) based on recommendation No. 2122/IPH.1/KS.02/X/2013 by the Indonesian Institute of Sciences (LIPI), and export permit SK.61/KSDAE/SET/KSA.2/3/2019 issued by the Directorate General of Nature Resources and Ecosystem Conservation (KSDAE) based on LIPI recommendation B-1885/IPH.1/KS.02.04/ VII/2017.

## Identification

From the three collected samples per plot, only the first two collected samples per plot were chosen to form the basis of this study due to the immense workload of morphological spider identification. Spiders from the first two samples per plot were identified to family and, if possible, to genus and morphospecies level using available literature (*Jocqué & Dippenaar-Schoeman, 2007*; *Murphy & Roberts, 2015*; *Deeleman-Reinhold, 2001*; *Koh & Bay, 2019*), the World Spider Catalog (https://wsc.nmbe.ch/) and the arachnological reference collections at the Zoological Museum in Hamburg (ZMH)). All spider morphospecies are documented pictorially in the Araneae section of the Ecotaxonomy database (http://ecotaxonomy.org/taxa/424669). The samples forming the basis of our study are continued to be used as reference material to identify further spider collections within the EFForTS project. Upon completion of spider identification, a collection of reference material will be deposited at the Museum Zoologicum Bogoriense at the Indonesian Institute of Science, LIPI.

## Biomass calculation

We measured the body length and body width of 15 randomly selected spider individuals per plot, including juveniles, to the nearest tenth of a millimeter using a ZEISS Stemi 2000 with fitted micrometer. The average spider body length and width per plot was used to calculate individual spider body mass based on taxon-specific allometric regression for tropical spiders (*Sohlström et al., 2018*), and the combined abundance of all spiders per square meter per plot was used to calculate total spider biomass per square meter per plot. All calculations, equations and raw data related to canopy spider biomass are given in the Supplements and the online data repository GRO (see data availability statement).

## Environmental variables

A set of environmental variables measured in the framework of the EFForTS project (EFForTS: Ecological and Socioeconomic Functions of Tropical Lowland Rainforest Transformation Systems; https://www.uni-goettingen.de/de/310995.html; *Drescher et al., 2016*) was used to explain canopy spider community composition in the four land-use systems. Measured in each plot, these variables included (1) mean canopy air temperature (°C) and (2) mean relative humidity (%), measured daily with a thermohygrometer (Galltec Mela, Bondorf, Germany) at 2 m height between April 2013 to March 2016 (*Meijide et al., 2018*), (3) canopy openness (%), measured with a spherical densitometer four times in each plot and then used as one average value (*Drescher et al., 2016*), (4) aboveground tree biomass [Mg/ha], calculated using diameter of trees, palms and lianas with diameter at breast height ≥ 10 cm and an allometric equation (*Kotowska et al., 2015*), (5) vascular plant species richness and (6) tree density based on 5 m × 5 m sub-plots where all tress with a diameter at breast height ≥ 10 cm were measured and identified [N/ha] (*Rembold et al., 2017*) and (7) mean stand structural complexity index, based on a Focus terrestrial laser scanner (Faro Technologies Inc., Lake Mary, USA) on a tripod at 1.3 m height (SSC; *Zemp et al., 2019*). Canonical Correspondence Analysis (CCA) was used to visualize the influence of environmental variables on canopy spider communities at both the morphospecies and family level. CCA was performed using vegan (*Oksanen et al., 2019*) in R (*R Core Team, 2019*). The final model was constructed using forward selection (vegan::ordir2step, direction = forward, permutations = 999) from the above environmental variables and community data. $R^2$ and variance partitioning were adjusted (*Borcard, Gillet & Legendre, 2018*) for the number of explanatory variables (vegan::RsquareAdj). CCA and forward selection were done separately for family and morphospecies community matrices.

## Statistical analyses

Statistical analyses were performed using R (v. 3.6.2., *R Core Team, 2019*) and visualized using ggplot2 (*Wickham, 2016*). Rank abundance curves were compared (vegan::radfit) and plotted (*Hartke, 2019*; https://github.com/tamarahartke/RankAbund) An exploratory data analysis was performed to ensure the data met underlying assumptions of the statistical tests (*Zuur, Ieno & Elphick, 2010*). The response variables abundance and biomass were analyzed using a generalized linear model (glm) with Gaussian error distribution and log link function (stats::glm). Response variables morphospecies richness and inverse Simpson Index 1/D (calculated using vegan::diversity; *Oksanen et al., 2019*) were analyzed using linear models (stats::lm). Initial models for all response variables included land use (rainforest, jungle rubber, rubber, oil palm), landscape (Bukit Duabelas, Harapan), and their interaction as fixed factors. Models were simplified in a stepwise manner discarding factors which did not significantly improve the fit of the model to find the minimal adequate model for each response variable. Model fit was checked using DHARMa (*Hartig, 2022*) after which multiple comparisons were made using pairwise *t*-tests with Holm corrections (multcomp::glht; *Hothorn, Bretz & Westfall, 2008*). Beta diversity was partitioned into turnover, nestedness and overall beta diversity using Sørensen pairwise dissimilarities (*Baselga et al., 2018*). Each partition was used for non-metric multidimensional scaling

(NMDS, vegan::metaMDS), and multivariate analysis of variance (MANOVA, Wilk's lambda) was used to test how well land use and landscape predicted the variability in NMDS scores; pair-wise contrasts were false discovery rate adjusted (*Benjamini & Hochberg, 1995*).

## RESULTS

In total, we collected 10,679 spider individuals from 32 research plots across four land-use systems. Of these, we determined 7,786 adult and subadult individuals to 36 families and 445 morphospecies (images of canopy spider families in Fig. S4, 1–36). Not all individuals could be determined to genus due to lack of relevant identification literature and a high proportion of undescribed species in putative new genera. Subadult individuals without fully developed sexual organs are usually not covered in identification keys, but we matched them with identified morphospecies based on general morphology whenever possible. The remaining 2,893 individuals could not be assigned to morphospecies because they were juveniles, however, they were determined to family based on general diagnostic features and thus included in the abundance (and biomass) analysis.

Overall, almost half of the specimens belonged to only four spider families (Salticidae, 2,043; Oonopidae, 1,878; Theridiidae, 1,533; Clubionidae, 1,188). Similarly, four families contributed 57.8% of all morphospecies: (Salticidae, 87; Theridiidae, 83; Araneidae, 48; Thomisidae, 39). More than half of all spider families comprised less than five morphospecies and less than 10% of all specimens identified. Of the 445 morphospecies recorded, 72 were exclusively found in the Bukit Duabelas landscape and 100 exclusively in the Harapan landscape (Fig. S5A). A total of 199 morphospecies (45%) were exclusively found in lowland rainforest and jungle rubber, while only 54 morphospecies (12%) were exclusively found in monoculture plantations of rubber or oil palm (Fig. S5B).

### Abundance, biomass and alpha diversity

When ranked by abundances, the number of canopy spider morphospecies and their abundances were lower in monocultures of rubber and oil palm than in rainforest and jungle rubber (Fig. 1). The models describing the shapes of the curves in the Whittaker plots significantly differed between rainforest and jungle rubber on one hand, and monocultures of rubber and oil palm on the other (Tukey's HSD, all four $T < -2.6$, $P < 0.03$). On average, canopy spiders in rainforest and jungle rubber were almost twice as abundant as in rubber plantations, and almost three times as abundant as in oil palm plantations, with the effect of land use being highly significant (glm; $F_{3,27} = 14.8$, $P < 0.001$; Fig. 2). Landscape also significantly affected canopy spider abundance (glm; $F_{1,26} = 7.1$, $P = 0.01$), but there was no significant interaction between the factors land use and landscape. Similar to abundance, canopy spider biomass was significantly affected by land use (glm; $F_{3,28} = 8.2$, $P < 0.001$), in that biomass in rainforest was more than twice as high as in rubber and almost four times as high as in oil palm, and biomass in jungle rubber intermediate (Fig. 3). Landscape did not significantly affect canopy spider biomass.

Canopy spider morphospecies richness was significantly affected by land use (glm; $F_{3,28} = 22.9$, $P < 0.001$) but not by landscape. On average, canopy spider morphospecies

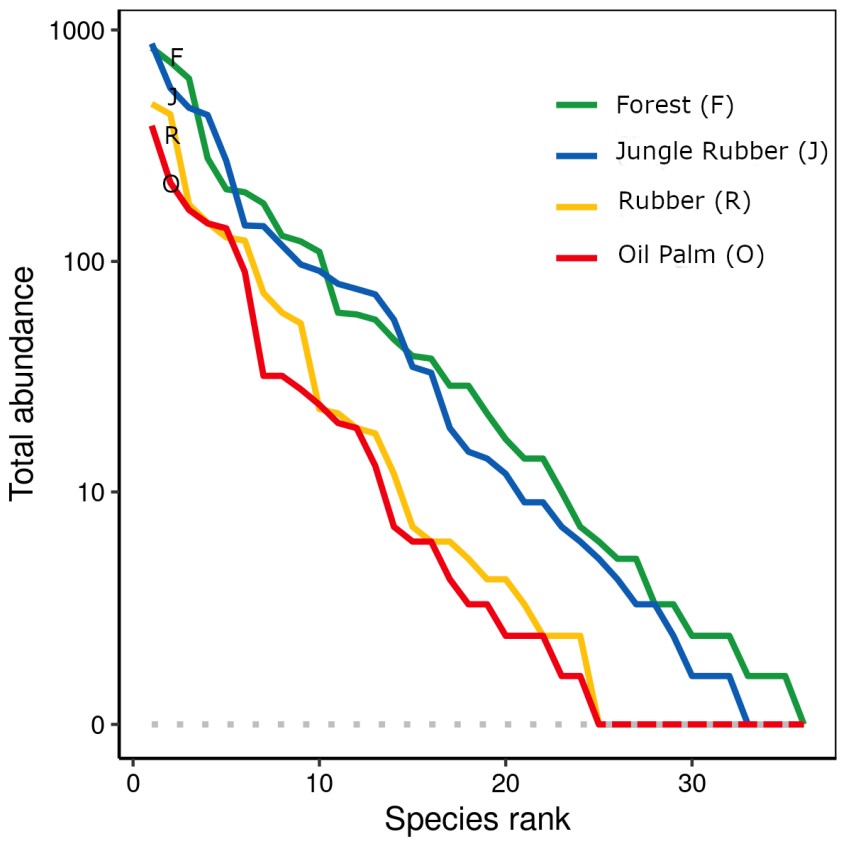

**Figure 1** **Rank abundance curves of 445 canopy spider morphospecies across four land-use systems in Jambi, Sumatra, Indonesia.** F, lowland rainforest; J, jungle rubber; R, rubber monoculture; O, oil palm plantation.

richness in rainforest (100.1 ± 21.4; mean ± *SD*) exceeded that in rubber (49.1 ± 11.4) and oil palm plantations (43.6 ± 10.7) by more than a factor of two, with jungle rubber being intermediate (87.9 ± 19.9; Fig. 4). By contrast, the inverse Simpson index was only marginally predicted by land use (glm; $F_{3,28} = 2.8$, $P = 0.06$) and not by landscape (glm; $F_{1,27} = 2.8$, $P > 0.09$).

## Community composition and beta diversity

The interaction between land use and landscape explained 82.1% of the total variance (Wilk's $\lambda = 0.179$, $F_{3,15} = 3.2$, $P < 0.001$) in canopy spider community composition, or overall beta diversity (land use: Wilk's $\lambda = 0.001$, $F_{3,15} = 40.8$, $P < 0.001$; landscape: Wilk's $\lambda = 0.163$, $F_{1,5} = 20.5$, $P < 0.001$). Overall, spider communities from rainforest and jungle rubber canopies were similar but differed from communities in rubber and oil palm monocultures, which in turn differed significantly from each other (Fig. 5). This pattern was mostly driven by turnover, which contributed almost the entire overall beta diversity, while nestedness contributed only marginally (Fig. 6). Consequently, an ordination of the two beta diversity partitions showed high resemblance of overall beta diversity with turnover (Fig. S6A), but not with nestedness (Fig. S6B). Both turnover and nestedness

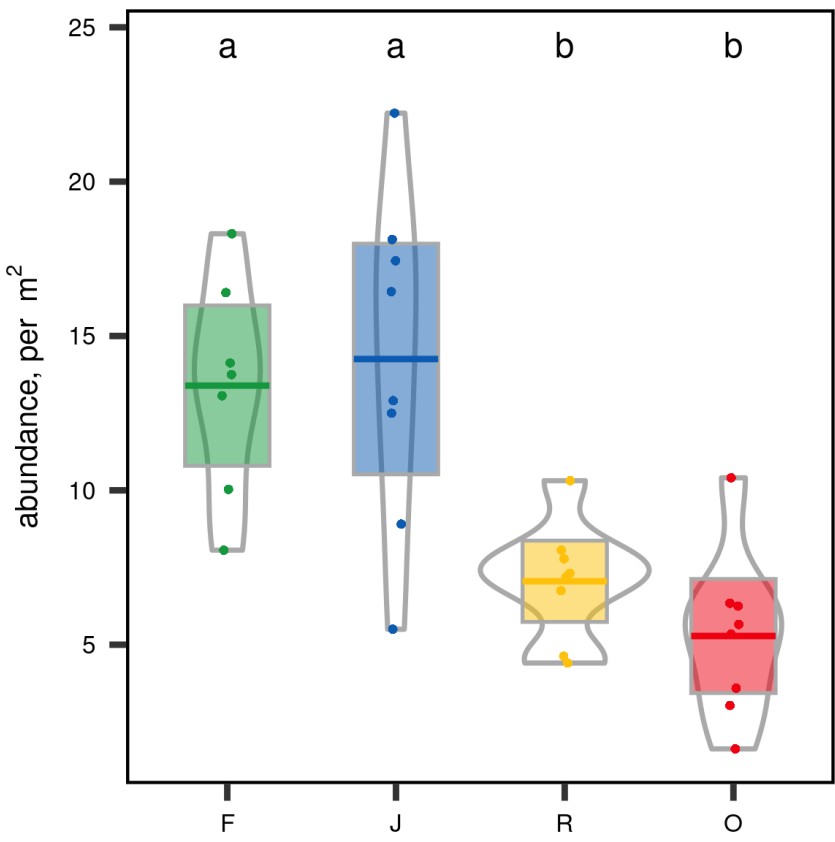

**Figure 2** **Canopy spider abundance [N/m$^2$] in four land-use systems in Jambi, Sumatra, Indonesia.** F, lowland rainforest; J, jungle rubber; R, rubber monoculture; O, oil palm plantation. Different letters indicate significant differences between land-use systems as indicated by Tukey's HSD ($P < 0.05$; dots = data points, bars = means, boxes = 95% C.I., violins = density).

overlapped between rainforest and jungle rubber, but were different from rubber and oil palm, which in turn overlapped.

## Influence of environmental variables

At family level, only three of the seven environmental variables significantly contributed to the model, canopy openness ($R^2_{adj} = 0.14$, $F = 5.95$, $P = 0.001$), aboveground biomass ($R^2_{adj} = 0.18$, $F = 2.63$, $P = 0.001$) and number of tree species per hectare ($R^2_{adj} = 0.21$, $F = 1.88$, $P = 0.012$). Increased canopy openness was associated with rubber and oil palm plantations, while trees per hectare and aboveground biomass were associated with jungle rubber and rainforest. The first three CCA axes (CCA1: $\chi^2 = 0.12$, $F = 6.80$, $P = 0.001$; CCA2: $\chi^2 = 0.05$, $F = 2.90$, $P = 0.003$; CCA3: $\chi^2 = 0.02$, $F = 1.35$, $P = 0.14$) together explained 20.8% of the variation in the data (CCA1 = 12.8%, CCA2 = 5.4%, CCA3 = 2.5%). Centroids of most canopy spider families clustered close to the center of the CCA graph and correlated little with the environmental variables, however Deinopidae and Selenopidae correlated closely with aboveground biomass and rainforest, and Liocranidae correlated closely with canopy openness and rubber and oil palm plantations (Fig. 7A). At

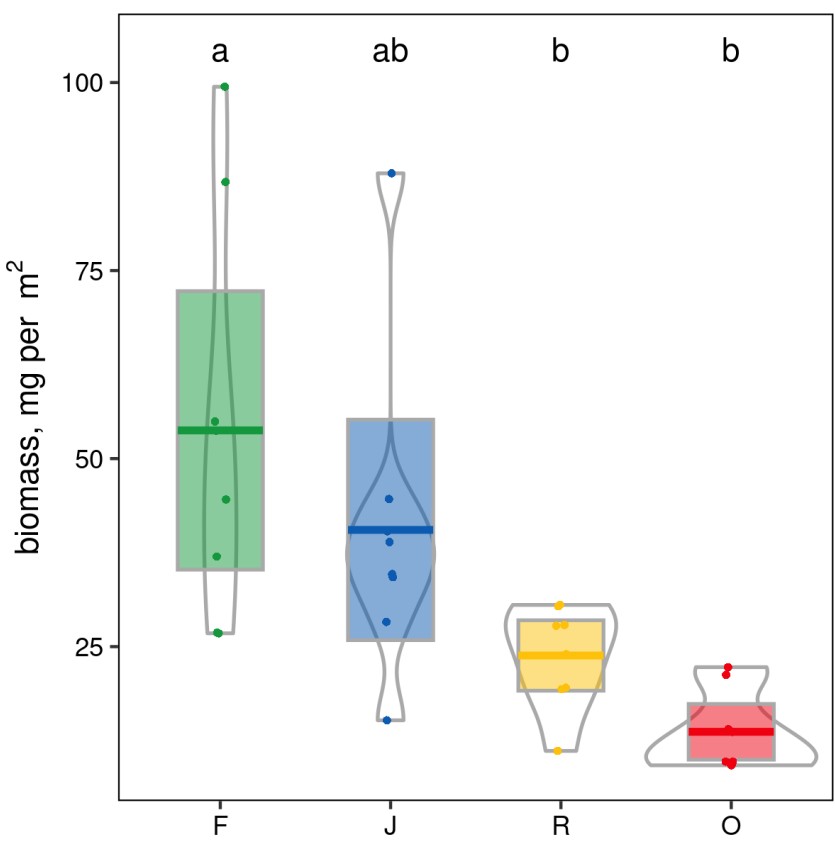

**Figure 3** **Pirate plots of canopy spider biomass in four land-use systems in Jambi, Sumatra, Indonesia.** F, lowland rainforest; J, jungle rubber; R, rubber monoculture; O, oil palm plantation. Different letters indicate significant differences between land-use systems as indicated by Tukey's HSD ($P < 0.05$; dots = data points, bars = means, boxes = 95% C.I., violins = density).

morphospecies level, only the environmental variables plant species richness ($R^2_{adj} = 0.05$, $F = 2.64$, $P = 0.001$) and number of tree species per hectare ($R^2_{adj} = 0.06$, $F = 1.54$, $P = 0.002$) significantly contributed to the model. The two CCA axes (CCA1: $\chi^2 = 0.47$, $F = 2.73$, $P = 0.001$; CCA2: $\chi^2 = 0.25$, $F = 1.49$, $P = 0.002$) together explained 6.8% of the variation in the data (CC1 = 4.4%, CCA2 = 2.4%). Similar to the family level CCA, most morphospecies clustered around the center of the ordination. The 39 morphospecies with scores >1.5 along the first axis belonged to the families Theridiidae (eight), Araneidae, Salticidae and Thomisidae (five each), Corinnidae and Uloboridae (three each), Gnaphosidae (two), and Clubionidae, Deinopidae, Linyphiidae, Liocranidae, Psechridae, Scytodidae, Sparassidae and Tetragnathidae (one each) (Fig. 7B). The greatest number of morphospecies was associated with rainforest and jungle rubber, few with rubber plantations, and none with oil palm plantations.

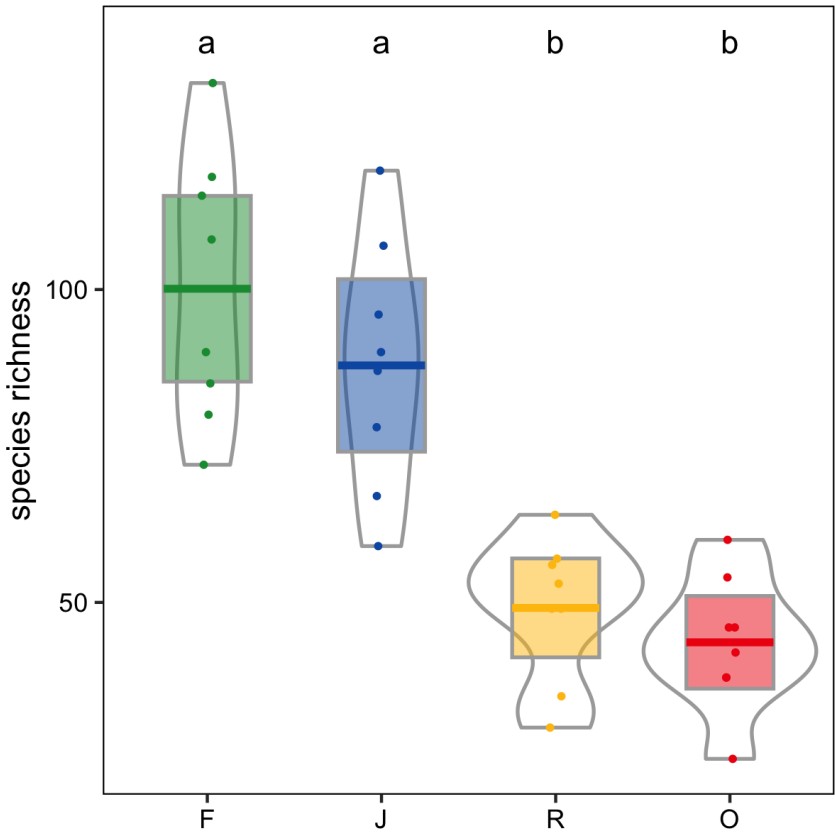

**Figure 4** **Canopy spider morphospecies richness in four land-use systems in Jambi, Sumatra, Indonesia.** F, lowland rainforest; J, jungle rubber; R, rubber monoculture; O, oil palm plantation. Dfferent letters indicate significant differences between land-use systems as indicated by Tukey's HSD ($P < 0.05$; dots = data points, bars = means, boxes = 95% C.I., violins = density).

## DISCUSSION

We investigated the effect of lowland rainforest conversion into jungle rubber, rubber and oil palm monoculture plantations on abundance, biomass, richness and community composition of canopy spiders in Sumatra, Indonesia. The study provided novel insight into the responses of one of the most important invertebrate predators to the transformation of lowland rainforest into agroforest systems and intensively managed monoculture plantations in one of the least studied biodiversity hotspots on this planet, the tropical region of Southeast Asia (*Myers et al., 2000*).

### Abundance, biomass and alpha diversity

Abundance, biomass and morphospecies richness in plantations of rubber and oil palm were significantly lower than in rainforest and jungle rubber, confirming our first hypothesis and supporting previous studies on arthropod diversity in these land uses, including ants (*Nazarreta et al., 2020*; *Kreider et al., 2021*), butterflies (*Panjaitan et al., 2020*), salticid spiders (*Junggebauer et al., 2021*) and parasitoid wasps (*Azhar et al., 2022*). The decrease in canopy spider abundance from rainforest to plantation systems also parallels findings of

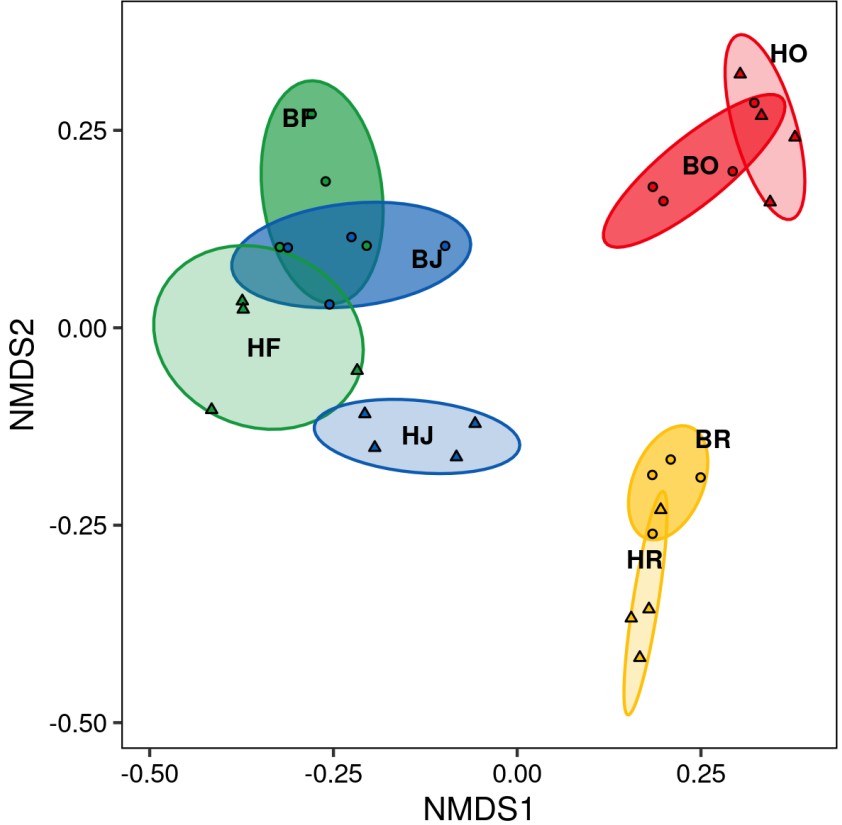

**Figure 5** **Non-metric multi-dimensional scaling (NMDS) of canopy spider community composition (= overall beta diversity) in four land-use systems and two landscapes in Jambi, Sumatra.** green = rainforest, blue = jungle rubber, yellow = rubber and red = oil palm; B, Bukit Duabelas landscape; H, Harapan landscape; F, lowland rainforest; J, jungle rubber; R, rubber monoculture; O, oil palm plantation.

earlier studies on canopy spiders in other biomes, such as old vs. young forests in eastern Europe (*Otto & Floren, 2007*) and secondary forest vs. rubber plantations in southwest China (*Zheng & Yang, 2015*). However, the average abundance of canopy spiders varies wildly between the few studies available. We collected 15.1 ind. m$^{-2}$ (all individuals) and 10.8 ind. m$^{-2}$ (identified individuals) in lowland rainforest in Sumatra. By comparison, between 0.97 and 14.6 ind. m$^{-2}$ of canopy spiders were sampled in old-growth rainforests in Sulawesi (*Russel-Smith & Stork, 1994*), 5.8 ind. m$^{-2}$ in montane forests in Tanzania (*Sørensen, 2004*) and ca. 30 ind. m$^{-2}$ in secondary forests in southwest China (*Zheng & Yang, 2015*). The differences might be due to different fogging methods, but likely also reflect different densities of canopy spiders in various forests across the tropical/subtropical zone. The uniform decline in the abundance of canopy spiders with the conversion of forest into plantation systems reported by *Zheng & Yang (2015)* and in our study indicates increased risk of local extinction of spider species in plantations (*Ceballos, Ehrlich & Dirzo, 2017*; *Hallmann et al., 2017*; *Sánchez-Bayo & Wyckhuys, 2019*). This may compromise the role of spiders as antagonists of herbivore prey species, ultimately threatening ecosystem functioning (*Soliveres et al., 2016*; *Dislich et al., 2017*).

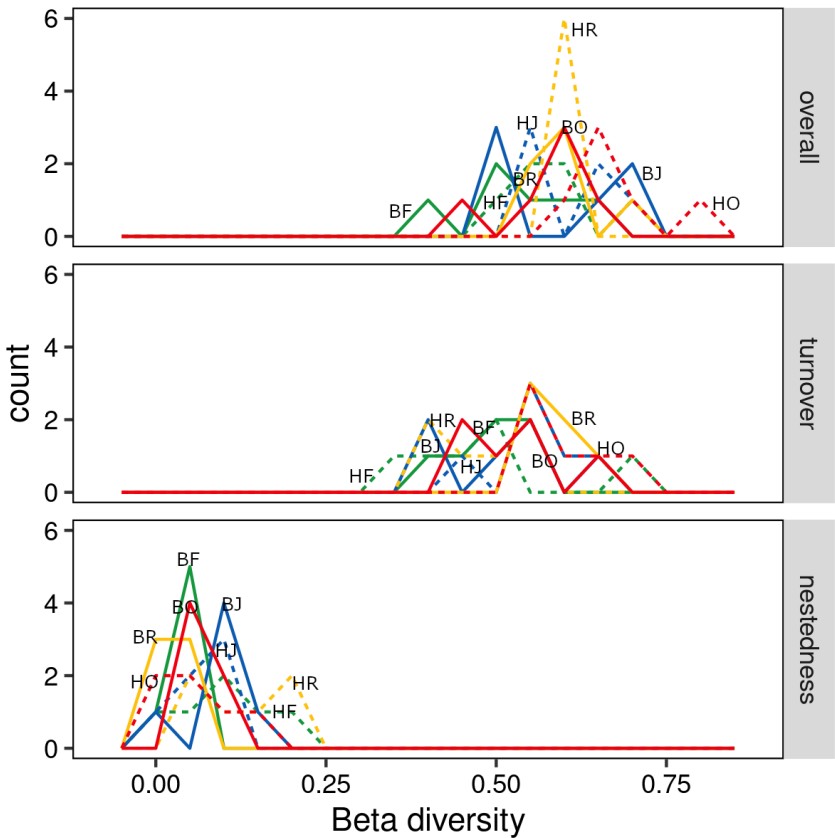

**Figure 6** **Overall beta diversity of canopy spiders (top) and the relative contributions of its partitions turnover (mid) and nestedness (bottom) in four land-use systems.** (green = rainforest, blue = jungle rubber, yellow = rubber and red = oil palm; B, Bukit Duabelas landscape; H, Harapan landscape; F, Lowland Rainforest; J, Jungle Rubber; R, Rubber monoculture; O, oilpalm plantation).

Canopy spider biomass differences between land-use systems mirror the differences of abundances between the land-use systems, and suggest that the contribution of spiders to ecosystem functions and services in rainforest and jungle rubber are at least twice as high as in monocultures of rubber and oil palm (*Boudreau, Dickie & Kerr, 1991*; *Barnes et al. , 2017*; *Sohlström et al., 2018*). This likely is related to the fact that total aboveground tree biomass in rainforest is more than twice that in jungle rubber, and more than four times that in rubber and oil palm (*Kotowska et al., 2015*).

Canopy spider morphospecies richness in rubber and oil palm plantations was less than half that in rainforest. This loss in morphospecies richness with the conversion of rainforest into monoculture plantation systems is similar to patterns reported from southeast China, where rubber plantations had 42.6–50.0% fewer canopy spider morphospecies than secondary forest (*Zheng & Yang, 2015*). Similar differences have also been found between natural and young managed forests in Europe (*Otto & Floren, 2007*). Overall, our data provide further support that conversion of natural or secondary forests into agricultural systems results in strong losses of species and overall biodiversity decline (*Sala et al., 2000*;

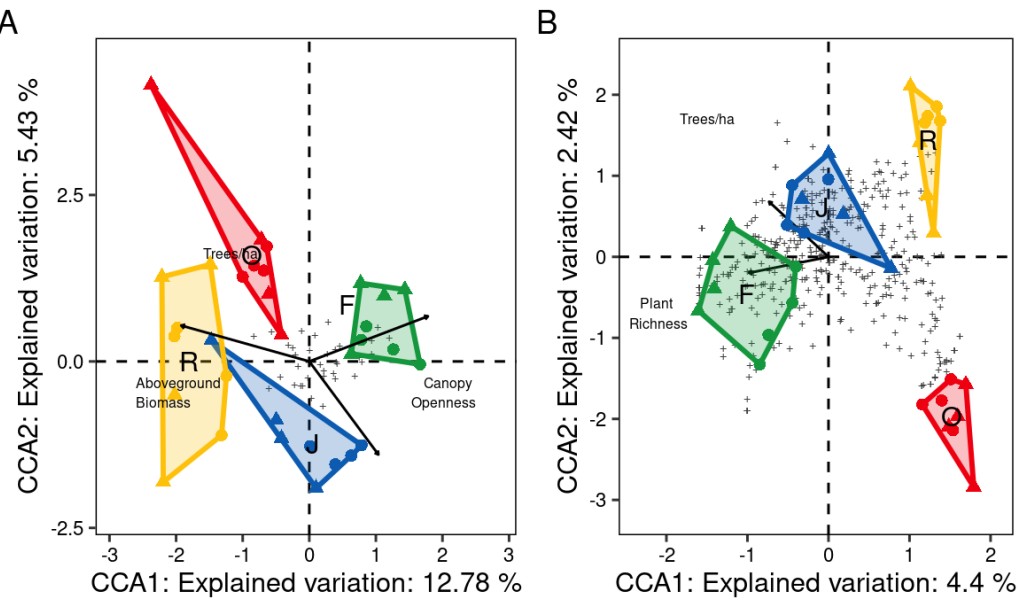

**Figure 7** Canonical Correspondence Analysis (CCA) of canopy spider community composition of four tropical land-use systems in Jambi, Sumatra, Indonesia. (A) Family level ($N = 36$) and (B) morphospecies level ($N = 446$) (F, rainforest; J, jungle rubber; R, rubber; O, oil palm). Only environmental factors significantly contributing to spider community composition are given. Plus symbols are individual families in (A), and individual morphospecies in (B).

*Sodhi et al., 2004*; *Steffan-Dewenter et al., 2007*; *Mumme et al., 2015*; *Newbold et al., 2015*; *Grass et al., 2020*; *Potapov et al., 2020*).

   Spider species in monoculture plantations were a subset of those found in rainforest, comprising species resilient against the transformation process and the changed environmental conditions in plantations. These findings are in line with earlier studies on other arthropod groups including canopy ants (Hymenoptera: Formicidae) (*Nazarreta et al., 2020*; *Kreider et al., 2021*), butterflies (*Panjaitan et al., 2020*), salticid spiders (*Junggebauer et al., 2021*) and parasitoid wasps (*Azhar et al., 2022*). *Nazarreta et al. (2020)* found that the conversion of rainforest into jungle rubber results in moderate species loss, suggesting that the majority of canopy ant species are resilient against moderate changes in land use. Presumably, the same is true for spider species of certain families, *e.g.*, Salticidae, Theridiidae and Oonopidae, which reach similar diversity in rainforest and jungle rubber. The strong decline in the richness of spiders, as well as other canopy arthropod taxa such as ants (*Nazarreta et al., 2020*), with conversion of rainforest into monoculture plantations of rubber and oil palm suggests that intensification of land use may critically compromise ecosystem functions and services provided by canopy arthropod predators and omnivores (*Power, 2010*; *Junggebauer et al., 2021*).

## Community composition and beta diversity
Similar to abundance and species richness, canopy spider community composition was affected by land use and landscape. Rainforest and jungle rubber communities were similar,
but differed strongly from those of oil palm and rubber plantations, confirming our second hypothesis. Shifts in community composition associated with land-use changes have been investigated in a wide range of tropical arthropods including ground spiders (*Potapov et al., 2020*), jumping spiders (*Junggebauer et al., 2021*), ants (*Nazarreta et al., 2020*; *Kreider et al., 2021*), butterflies (*Panjaitan et al., 2020*), pseudoscorpions (*Liebke et al., 2021*), salticid spiders (*Junggebauer et al., 2021*) and parasitoid wasps (*Azhar et al., 2022*). Generally, within in each of these taxa, a number of generalist species tolerate increased disturbance and the harsher environmental conditions in plantations. For spiders this suggests that certain species tolerate disturbances in plantations as long as essential habitat requirements are met, such as structural elements to allow attachment of webs (*Halaj, Ross & Moldenke, 2000*; *Jiménez-Valverde & Lobo, 2007*; *Ávila et al., 2017*; *Ganser et al., 2017*; *Rao, 2017*).

By contrast, a range of spiders predominantly occur in tropical rainforests compared to a variety of disturbed habitats as shown for *Aporosa yunnanensis* forests compared to rubber-tea mixture and rubber plantations (*Zheng et al., 2017*), firewood plantations compared to grasslands and cultivated wetlands (*Chen & Tso, 2004*), old growth forests compared to younger re-forested areas (*Floren & Linsenmair, 2001*; *Floren & Deeleman-Reinhold, 2005*) and rainforest compared to rubber and oil palm plantations (*Potapov et al., 2019*; *Potapov et al., 2020*). Although sampling methodologies differed between these studies, only few families, including web-building Araneidae, Theridiidae and Tetragnathidae, and the free hunting Corinnidae, Salticidae, Sparassidae and Thomisidae, contributed most to overall spider abundance and species richness (61–94%). Similarly, these families also contributed most to total abundance (57%) and richness (59%) of spiders in rainforest in our study, and even more to the overall abundance (63% and 68%) and richness (74% and 72%) in rubber and oil palm plantations. Differences in relative abundance and species richness between rainforest and plantations indicate different sensitivities of spider families to disturbance. A total of 14 families, including Anapidae, Ctenidae and Deinopidae, were present in our rainforest samples but absent in rubber and oil palm plantations, suggesting that species of these families are particularly sensitive to disturbance and the altered abiotic and biotic conditions in plantation systems.

## Influence of environmental variables

Three of the seven studied environmental variables affected the spider community structure at family or morphospecies level, supporting our third hypothesis. At the family level, changes in canopy openness, aboveground tree biomass and tree density contributed to the shift in spider communities from rainforest to monoculture plantations. Canopy structure and tree diversity have been identified previously as drivers of canopy spider communities. *Floren & Deeleman-Reinhold (2005)* found reduced spider diversity in disturbed isolated forest patches with more open canopies compared to less disturbed regenerated forest, while *Jiménez-Valverde & Lobo (2007)* found that richness of orb-weavers (Araneidae) and crab spiders (Thomisidae) correlate closely with shrub canopy and ground herb cover, *i.e.*, vegetation complexity. In our study, canopy openness was closely associated with oil palm and rubber plantations, which are generally characterized by lower canopy complexity compared to rainforest and jungle rubber (*Zheng & Yang, 2015*; *Drescher et*
*al., 2016*; *Zemp et al., 2019*). Our results indicate that reduced complexity detrimentally affected a wide range of spider taxa, but may also favor specialist species benefitting from associated increase in temperature and light, *e.g.*, by facilitating hunting of prey via optical cues (*Fayle et al., 2010*; *Ganser et al., 2017*). In fact Liocranidae, predominantly comprising surface-hunting species known to prefer open habitats (*Deeleman-Reinhold, 2001*), flourished in plantations.

Aboveground biomass, which increases with plant species richness, vegetation cover, height and age of trees (*Vogel et al., 2019*), was identified as predictor for spider community composition in our study. Similar studies have found spiders communities to be negatively affected by low tree density (*Barton et al., 2017*) and to benefit from high tree species richness and height (*Schuldt et al., 2011*). Further, *Floren et al. (2011)* found spider communities in southeast Asia to benefit from tree age, suggesting that older trees support a wider range of spider species. Results of our study indicate that in particular the net-casting Deinopidae and the ambush hunting Selenopidae, which only occurred in rainforest, benefit from high aboveground biomass as also suggested by *Deeleman-Reinhold (2001)* and *Floren et al. (2011)*. Potentially, the specific hunting technique used by these spiders combined with a greater degree of habitat specialization contributed to their high sensitivity to rainforest conversion.

At the morphospecies level, spider community structure only correlated closely with plant diversity-associated variables (plant species richness and number of tree species per hectare) and the variation in species distribution was not well explained by the studied environmental variables (combined explanatory power of first two CCA axes 6.9%, compared to 18.2% at family level). Presumably, stochastic processes play a more pronounced role in structuring spider communities at morphospecies level than at the level of families. The close correlation with plant diversity-associated variables likely reflects the fact that habitat preferences at morphospecies level are more specific than at family level. Plant species richness is known to be an important driver of predator arthropods such as ants or spiders in both temperate and tropical forest ecosystems (*Schuldt et al., 2011*; *Drescher et al., 2016*; *Matevski & Schuldt, 2021*). *Samu et al. (2014)* found 26% of the variation in spider assemblages to be explained by tree species composition and showed certain spider species to be associated with specific tree species in temperate forests. Similarly, *Schuldt et al. (2011)* found certain spiders species to be associated with individual tree species even in forests with high tree diversity. Canopy spider diversity also has been found to closely correlate with vegetation complexity and other plant variables in tropical forests (*Zheng & Yang, 2015*). Despite being the most prominent variables explaining spider community composition at morphospecies level, plant diversity-associated variables only explained a small proportion of the variability in our spider communities suggesting that other factors are likely to be more important for structuring spider communities at species level. In addition to other environmental variables, interactions with other species, including prey and predators, inter-specific competition and intra-guild interactions, may contribute to the local assemblage of spider species (*Sih, Englund & Wooster, 1998*; *Mooney, 2007*; *Mestre et al., 2013*). Elucidating the role of these interactions for canopy spider community

composition is difficult to infer and requires experimental studies in the field, which are difficult to establish in tropical forest ecosystems.

## CONCLUSIONS

Overall, the results showed that canopy spider communities in oil palm and rubber plantations are less abundant, contain lower biomass and are less diverse compared to the more natural ecosystems rainforest and jungle rubber. Notably, species composition of spider communities was similar in rainforest and jungle rubber, and differed strongly from that in oil palm and rubber plantations, with the latter also differing from one other. At family level, aboveground biomass, number of trees per hectare and canopy openness were identified as major environmental factors determining spider community composition, while at species level the most important factors were plant richness and number of trees per hectare. The results highlight the importance of rainforest for the conservation of canopy spider communities, as only a subset of the community can tolerate the harsh environmental conditions and disturbances in monoculture plantations. Similar diversity and community composition in rainforest and jungle rubber highlights that the majority of spiders tolerates moderate disturbances and decline in trees species indicating that agroforest systems may contribute substantially to the conservation of tropical canopy spider communities.

## ACKNOWLEDGEMENTS

Nop, Yohanes Bayu Suharto, Yohanes Toni Rohaditomo, and Zulfi Kamal helped during sample collection. Clara D. Zemp, Dominik Seidel, Martin Ehbrecht, Holger Kreft and Dirk Hölscher provided Stand Structural Complexity Index data on plot level. The comments of three reviewers improved an earlier version of the manuscript. We thank village leaders, local plot owners, PT Humusindo, PT REKI, PT Perkebunan Nusantara VI, and Bukit Duabelas National Park for granting us access to and use of their properties.

### Funding

This study was funded by the Deutsche Forschungsgemeinschaft (DFG, German Research Foundation) project number 192626868 in the framework of the collaborative German - Indonesian research project CRC990 - EFForTS. Daniel Ramos was funded by the Katholische Akademische Austauschdienst KAAD. The publication of this study was supported by the Open Access Funds of the Göttingen University. The funders had no role in study design, data collection and analysis, decision to publish, or preparation of the manuscript.

### Grant Disclosures

The following grant information was disclosed by the authors:
DFG, German Research Foundation:  192626868.

German - Indonesian Research project: CRC990 - EFForTS.
Katholische Akademische Austauschdienst KAAD.
Open Access Funds of the Göttingen University.

## Competing Interests

The authors declare there are no competing interests.

## Author Contributions

- Daniel Ramos conceived and designed the experiments, performed the experiments, analyzed the data, prepared figures and/or tables, authored or reviewed drafts of the article, and approved the final draft.
- Tamara R. Hartke conceived and designed the experiments, analyzed the data, prepared figures and/or tables, authored or reviewed drafts of the article, and approved the final draft.
- Damayanti Buchori conceived and designed the experiments, authored or reviewed drafts of the article, and approved the final draft.
- Nadine Dupérré performed the experiments, authored or reviewed drafts of the article, and approved the final draft.
- Purnama Hidayat conceived and designed the experiments, authored or reviewed drafts of the article, and approved the final draft.
- Mayanda Lia performed the experiments, authored or reviewed drafts of the article, and approved the final draft.
- Danilo Harms performed the experiments, authored or reviewed drafts of the article, and approved the final draft.
- Stefan Scheu conceived and designed the experiments, analyzed the data, prepared figures and/or tables, authored or reviewed drafts of the article, and approved the final draft.
- Jochen Drescher conceived and designed the experiments, performed the experiments, analyzed the data, prepared figures and/or tables, authored or reviewed drafts of the article, and approved the final draft.

## Field Study Permissions

The following information was supplied relating to field study approvals (i.e., approving body and any reference numbers):

The samples of canopy arthropods on sites of farmers in Jambi Province, Sumatra, Indonesia, in the framework of the Collaborative Research Centre 990 / EFForTS are taken in consent with the farmers. Access to their sites and taking of samples is based on individual agreements between the EFForTS partner University UNJA, the University of Jambi, and the owners of the farmland. In total 35 contracts have been signed and might be accessed on request.

Access to field sites in Harapan Rainforest was granted through a bilateral agreement between UNJA and PT REKI (Perseroan Terbatas Restorasi Ekosistem Indonesia) which manages Harapan Rainforest. Access to field sites in Bukit Duabelas National Park was granted through a bilateral agreement between UNJA and the National Park administration.

The basis of the contracts is the Memorandum of Agreement (MoA) between the Consortium of IPB University, University of Jambi and Tadulako University and the University of Göttingen concerning the Collaborative Research Centre 990 / EFForTS.

The study was conducted using samples/organisms collected based on Collection Permit No. S.710/KKH-2/2013 issued by the Ministry of Forestry (PHKA) based on recommendation No. 2122/IPH.1/KS.02/X/2013 by the Indonesian Institute of Sciences (LIPI), and export permit SK.61/KSDAE/SET/KSA.2/3/2019 issued by the Directorate General of Nature Resources and Ecosystem Conservation (KSDAE) based on LIPI recommendation B-1885/IPH.1/KS.02.04/ VII/2017.

## Data Availability

The data that support the findings of this study are openly available on the GRO Göttingen Research Online data repository: https://data.goettingen-research-online.de/ and under the following DOI's:

(1) Abundance by morphospecies:

Drescher, Jochen, 2022, "Ramos.2022.PeerJ_Canopy.Spiders.2013_Abundance by morphospecies", https://doi.org/10.25625/L17S8R, GRO.data, V2, UNF:6:K734wr5nl3I/hcvdnKeFrw== [fileUNF]

(2) Abundance by family:

Drescher, Jochen, 2022, "Ramos.2022.PeerJ_Canopy.Spiders.2013_Abundance by family", https://doi.org/10.25625/TOIEX0, GRO.data, V1, UNF:6:46qYx0u2DoNhNwBzR1NkVg== [fileUNF]

(3) Body length/width, biomass:

Drescher, Jochen, 2022, "Ramos.2022.PeerJ_Canopy.Spiders.2013_BodySizes.Biomass", https://doi.org/10.25625/QAFDUM, GRO.data, V1, UNF:6:xGfS3nPyL5rE3IvApqZQGA== [fileUNF]

(4) Environmental variables:

Drescher, Jochen, 2022, "Ramos.2022.PeerJ_Environmental.Variables.2013", https://doi.org/10.25625/DPPATX, GRO.data, V1, UNF:6:fEW8PcSOlbSEiDognFVjzg== [fileUNF]

The R script used for the analysis is available in the Supplementary Files.

## Supplemental Information

Supplemental information for this article can be found online at http://dx.doi.org/10.7717/peerj.13898#supplemental-information.

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
