# Peer review of "Rainforest conversion to rubber and oil palm reduces abundance, biomass and diversity of canopy spiders"

_PeerJ, doi:10.7717/peerj.13898_

## Round 0.1 · original submission · Minor Revisions

Dear Authors

Thank you for submitting the manuscript to PeerJ. Now the reviews on your manuscript have been received. Changes are required in the manuscript before consideration, particularly against the comments of Reviewer 1 and Reviewer 3.

·

Basic reporting

This article represents the culmination of a great deal of fieldwork, curation and ID, and data analyses.
It presents clear hypotheses which are tested against the data and ties the results to relevant literature, and as such represents a valuable contribution to the literature on spider diversity under differing land-use conditions.
The figures are clear and informative, and I trust that the final versions will be reproduced in a better quality than the automated PDF generation of PeerJ allows.

The raw data is available for download, but I would urge the authors to share their code, including the package versions and base R installation information so that readers can reproduce the analyses for critique and/or learning.

More detail in the methods for describing and quantifying habitat parameters should be included, so that reproduction of the study can be easily attempted.The raw data underlying these parameters is included in the datasets, as currently the analysis cannot be reproduced with the datasets given.

If the specimens collected in this work were deposited (and they should be) in an accredited institutional collection, the authors should indicate this, and also tell the readers which collections received the deposits.

Experimental design

As noted above, the data on habitat descriptors forming inputs for the models should be included.
The article does a good job of laying out their questions, data, and conclusions, as well as the broader relevance of the conclusions.

Validity of the findings

Seems to fit in with the findings found in other major studies of this phenomenon, but I have no specific expertise in these types of diversity studies.

Additional comments

Congratulations on a well-thought-out and well executed study!!!

·

Basic reporting

The paper intitulate " Rainforest conversion to rubber and oil palm reduces abundaces and diersity of canopy spiders" compared to the abundance and diversity of families associated with treetops. The article is very well written, with excelent theoretical foudatiin and appropriate analysis

Experimental design

The experimental design is excelent and with well-defined hypotheses. Statisticalanalysesand methodologies are appropriate.

Validity of the findings

The results also are excelent and confirm the hypotheses proposed in thecstudy.

Additional comments

The article shows the importance of the consevstion of natural forest versus monoculture tree forest. The article will have a very good impact on international literature

·

Basic reporting

Ramos et. al. present work on the effects of rainforest conversion on the diversity of canopy spiders. Their results confirm that monoculture farming reduces diversity, but nonmonoculture practices such as jungle rubber production mitigate these effects to some degree. Overall, it is a well-conceived, well-written paper. Any comments I have are relatively superficial.

I have given my basic comments per line as numbered in the word document I received-

*Note- I have given some grammar suggestions (particularly concerning commas) that may vary by writing style and country of origin. Know that they are just my suggestions. If they are corrections you prefer then take them, if they are not the way in which you would write, then feel free to disregard.*

Line 51 – remove comma “data, but”
Line 58 – “aide” should be “aid” (aide refers to a person)
Lines 79-80 – I think siting BPS 2018 once at the end of the sentence is probably sufficient?
Line 81 – uncapitalize province
Line 102 – remove comma after beetles
Line 130 – agroforestry system in which rubber trees…
Line 131 – successively degrading or successively degraded?
Line 140 – two hours
Line 141 – for future use
Line 153 – define “EFForTS”
Line 230 – remove comma after ‘similar’
Lines 231 & 238 and Fig 5 – This line refers to figure 5a, but figure 5 does not have panels labeled by letter. I’m assuming that the left-hand side is 5a, but a label would be nice. It is also possible that this is explained in a figure legend, but I cannot find it. A figure legend would also explain what the colors and abbreviations mean. I assume there should be legends somewhere, I just can’t find them.
Lines 253 & 262 and Fig 6 – Same issue as above
Line 311 – Empty parenthesis/missing reference
Line 338 – huntering should be hunting

I was not able to find any figure legends for the main figures, only the supplementary material. I am assuming they exist and just didn't make it into the manuscript upload.

Experimental design

No comment

Validity of the findings

No comment

---

## Round 0.2 · accepted · Accept

Dear Authors

I am writing to inform you that your manuscript has been accepted for publication.